# Laser light sources for photobiomodulation: The role of power and beam characterization in treatment accuracy and reliability

**Carlos Eduardo Girasol**[1], **Guilherme de Araújo Braz**[2], **Luciano Bachmann**[3], **Jonathan Celli**[4], **Rinaldo Roberto de Jesus Guirro**[1]*

**1** Postgraduate Program in Rehabilitation and Functional Performance, Department of Health Sciences, Ribeirão Preto Medical School, University of São Paulo (USP), Ribeirão Preto, São Paulo, Brazil, **2** Technology Center of the Supera Park of Innovation and Technology of Ribeirão Preto, Ribeirão Preto, São Paulo, Brazil, **3** Postgraduate Program in Physics Applied to Medicine and Biology, University of São Paulo (USP), Ribeirão Preto, São Paulo, Brazil, **4** Department of Physics, University of Massachusetts Boston, Boston, Massachusetts, United States of America

* rguirro@fmrp.usp.br

**Data Availability Statement:** Zenodo - https://zenodo.org/record/5773138#.YbO0bb3MLIU doi: 10.5281/zenodo.5773138.

## Abstract

### Purpose

Daily clinical use of therapeutic light sources can lead to changes in light emission stability with potentially significant consequences for usage in photomedicine treatment. The aim of this study was to evaluate the average and maximum power and to describe the beam diameter of different low-power laser photobiomodulation devices in clinical use in Brazil.

### Methods

The power and light-emitting beam diameter of twenty-four therapeutic devices with an average age of 11±5 years, with an average weekly use of fewer than thirty minutes, were measured.

### Results

The analyzed power varied between 2% to 134% of the values declared by the manufacturers. Differences in beam diameter of between 38% and 543% of the nominal values were also observed. It is also noteworthy that even between the same brand and model, differences in diameter were obtained. Finally, differences were observed in the power output after one and three minutes of sequential emission for 830 nm and 904 nm ($p < 0.05$), but not when comparing the difference between wavelengths in factor time.

### Conclusion

There is a need for a shared effort on the part of laser manufacturers to improve standardization and consistency of laser output power and beam diameters. At the same time, medical laser operators should also consider development of standardized protocols for maintenance and monitoring equipment performance over time to correct for fluctuations that could ultimately impact on treatment outcomes.

**Funding:** CEG - The São Paulo Research Foundation - FAPESP (grant #2018/14955-6). RRJG - Coordination for the Improvement of Higher Education Personnel - Brazil (CAPES) – Financing Code 001. The funders had no role in study design, data collection and analysis, decision to publish, or preparation of the manuscript.

**Competing interests:** The authors have declared that no competing interests exist.

## Introduction

Photobiomodulation therapy has seen a period of extensive growth and broader adoption in a variety of clinical applications [1]. However, as already pointed out by Enwemeka [2], uncertainty in light dosimetry may contribute to incomplete biological response. To achieve a controlled and predictable biological response it is essential that light delivery parameters, starting with power and spectral properties of light sources, but also tissue optical properties, are well characterized [3,4].

While Brazil has emerged as a major contributor to the advancement of photobiomodulation research and clinical translation, studies regarding the preventive maintenance and condition of Brazilian equipment date back to more than a decade ago and have already highlighted that the maintenance of light-generating sources in clinical environments is being neglected. Guirro and Weis [5] conducted a study with equipment used in physical therapy clinics and pointed out that most of them presented power values below the nominal and declared values and even that some of them were not emitting light at all. Similarly, Fukuda and Malfatti [6] analyzed the final energy emitted by 904 nm and 905 nm equipment and obtained data conflicting with the information provided by the manufacturers.

In Brazil, the standards are defined in the Brazilian Committee of Dental-Medical-Hospital, together with the study commission of Electromedical Equipment, regulated in general by the Brazilian Association of Technical Standards (ABNT). Moreover, the Brazilian standard ABNT NBR IEC 60601-2-22:2014 [7,8] is an identical adoption of IEC 60601-2-22:2007 [9] prepared by the Subcommittee Optical Radiation Safety and Laser Equipment (SC 76). Thus, the need for knowledge of such standards stands out, both for professionals who handle the equipment for therapeutic purposes, as well as for developers, being pointed out as minimum requirements that need to be followed to achieve a reasonable level of safety and reliability during the operation and application of the equipment. A problem faced by the Brazilian scientific community and the developers is that the Brazilian standards are a direct translation of its international version and are not always well adapted for the existing technology in the country. Most of the knowledge necessary to understand and correctly apply the standard guidelines are retained in the academic community, and it is difficult for the developers to apply, with good understanding, the necessary tests and experiments to make sure their equipment follows the standard requirements.

Enwemeka [2] reports that trained professionals can only control the physical parameters of the laser to be used for each pathology, given its individuality. However, the equipment manufacturers have predefined some base fluence values without considering the individual characteristics of each subject. In this context, if the power and the beam diameter are different than expected due to a faulty function of the equipment, the quality of the treatment will be affected. Thus, the objective of this study was to evaluate and describe the laser beam diameter and the average power in different devices in clinical use in Brazil.

## Materials and methods

### Equipment

The equipment was obtained from physical therapy clinics, hospitals, universities, and research laboratories that use photobiomodulation equipment for therapeutic purposes. The inclusion criteria were according to the availability of laser devices in therapeutic use, with no exclusion according to brand, model, or physical parameterization.

Initially, the owners were informed about the research project, its objectives, and its characteristics. They were informed that it would be necessary to make the equipment available for

one day to the Laboratory of Physiotherapeutic Resources (LARF) of the Ribeirão Preto Medical School of the University of São Paulo (FMRP-USP). So, after the explanation and a verbal acceptance by the owner of the equipment were given, we proceeded to evaluate them. The equipment was not exposed to physical risks, nor were its owners exposed to moral damages or expenses. It is also important to note that no informed consent was required since we did not interact directly with humans or expose their individual information.

### Average power evaluation

To measure the average power of each light source, a PowerMax-USB Power Sensor (Coherent, Santa Clara, CA, USA) and its associated software were used. This system allowed the data acquisition on a laptop model RF511 (Samsung, Manaus, AM, Brazil).

The light source was positioned and mechanically stabilized to irradiate the power sensor directly at perpendicular incidence. The devices with wavelengths in the red spectrum were analyzed during one minute of sequential irradiation and the infrared ones during one and three minutes to observe possible emitted power changes. This high period for the 830 nm wavelength was defined by the question, still discussed in the Brazilian clinical environment, of a possible time required for stabilization of the power emitted by near-infrared wavelength devices available for local use. We emphasize that the power sensor had been calibrated by the manufacturing company since the acquisition time was less than 12 months. Thus, the error rate is between 1% and 1.5%, according to the producer.

### Beam diameter evaluation

The beam diameters of the laser emitters were collected using a visible radiation-sensitive detector (LM-2 VIS, Coherent, Santa Clara, CA, USA) or an infrared radiation-sensitive detector (LM-2 NIR, Coherent, Santa Clara, CA, USA), according to the laser equipment to be evaluated. The detector was connected to the power meter (FieldMaxII TOP, Coherent, Santa Clara, CA, USA) and equipped with an SMA (SubMiniature version A) optical adapter to hold a 5 cm segment of optical fiber with a 0.25 mm diameter. Thus, the sensor was coupled to an XYZ translation stage with 0.005 mm resolution to measure the power profile in the laser beam cross-section. The experimental setup can be visualized in Fig 1. The laser probe was

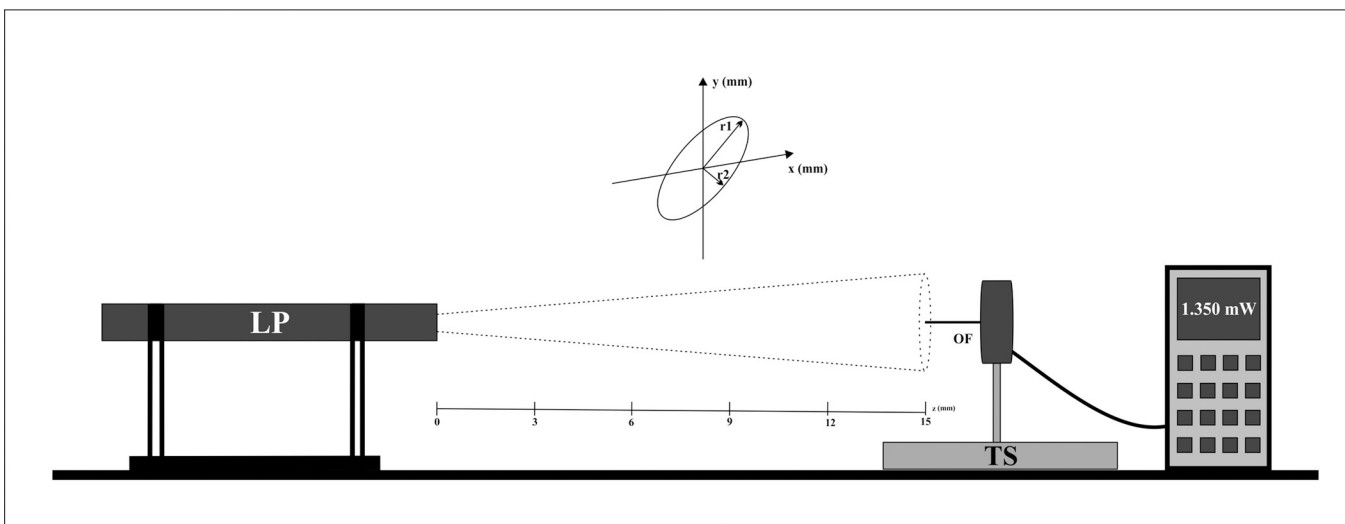

**Fig 1. Diagram of the experimental setup for the beam profile measurement.** LP: Laser probe; OF: Optical Fiber; TS: Translation Stage.

fixed, and the beam was aligned with the optical fiber coupled to the sensor. The optical fiber tip was positioned at the desired Z value (according to Fig 1 diagram), and the translation stage moved the fiber tip through (transversally) the laser beam measuring the power step by step. This measurement gave the beam power profile, i.e., the power density at Z-position, and was repeated at two cross-section directions (x-axis and y-axis). With this measurement, it was possible to determine the transversal profile of the beam to the Z = 0 position, the contact position; and was repeated for the Z = 3, 6, 9, 12, and 15-mm position.

For each cross-section measurement, we fit a Gaussian function to extract the beam radius, according to the equation below:

$$I(x) = I_b + I_0 exp\left[-0.5\left(\frac{x - x_c}{r}\right)^2\right] \qquad \text{Eq1}$$

where $I(x)$ is the power density, also called irradiance, $I_b$ the baseline offset, $I_0$ the peak irradiance at the center of the beam ($x_c$), and $r$ the desired radius of the beam is the radius.

The beam laser can have an elliptical cross-section. So, to an elliptical cross-section, Eq 2 was adopted to calculate the area of the beam in each Z position:

$$Ellipse\ Area = r_1 \times r_2 \times \pi \qquad \text{Eq2}$$

where the $r_1$ and $r_2$ are the two radii achieved by the Gaussian fit.

To evaluate the tridimensional profile of the beam for the different equipment, the measurement of the mean power was done, not only across the two-transversal x- and y-axes but covering the entire cross-section of the beam. This experimental procedure accesses the tridimensional profile of the beam. It is unnecessary to repeat this procedure for different Z values because the 3D profile is the same.

## Data processing and statistical analysis

For statistical analysis, the Shapiro–Wilk normality test was used to verify the data distribution. For comparison between the difference in mean and maximum power of the 904 nm devices, both for one and three minutes, a paired t-test was used. For comparison of the 830 nm devices between one and three minutes and compared against 904 nm, an unpaired Mann-Whitney test was used. It should also be noted that for the analysis of the power difference between the wavelengths, the difference emitted during 1 and 3 minutes was adopted, that is: $\Delta nm = P$ (3 $Min$)$-P$ (1 $Min$), where $\Delta nm$ is the wavelength of interest, and $P$ is the analyzed power.

Spearman's test was used to analyze the correlation coefficient between the equipment acquisition time and the last maintenance with the analyzed values of mean power and beam diameter. For all tests, a 5% significance level was considered. For data presentation, we opted to use percentages to describe the different powers, considering 100% the value declared by each equipment manufacturer. For the radius and area of the light source, the values were presented in the proper units. Data processing was performed in GraphPad Prism software version 7.0 (GraphPad Software, San Diego, CA, USA) and the Microcal Origin® software.

## Results

Twenty-four pieces of equipment were evaluated. All were in use in the physical therapy service routine before the evaluation was performed. All information can be found in Table 1 by the number of appearances for each event.

The average power fell within 90% to 110% of the predicted value from the manual for only nine devices. Overall, results spanning from 2% up to 134% of the quoted power were observed.

**Table 1. Characterization of the evaluated equipment (n = 24).**

| Equipment | Wavelength (nm) | Purchased time (months) | Last maintenance (months) | Average time of use/ week (minutes) | Average power measured (mW) | Percentage of the power predicted in manual |
|---|---|---|---|---|---|---|
| Ibramed—Laserpulse[1] | 904 | >120 | 12 to 24 | <30 | 41.9 | 60% |
| Ibramed—Laserpulse[1] | 450 | >120 | 12 to 24 | <30 | 62.37 | 89% |
| Ibramed–Laserpulse[1] | 660 | >120 | 12 to 24 | <30 | 33.06 | 110% |
| Ibramed–Laserpulse[1] | 830 | >120 | 12 to 24 | <30 | 33.06 | 110% |
| Ibramed—Laserpulse[1] | 830 | >120 | 12 to 24 | <30 | 0.61 | 2% |
| Ibramed–Laserpulse[1] | 830 | >120 | 12 to 24 | <30 | 40.28 | 134% |
| Ibramed—Laserpulse[1] | 830 | >120 | over 36 | <30 | 33.63 | 112% |
| Ibramed–Laserpulse[1] | 830 | >120 | over 36 | <30 | 33.77 | 113% |
| Ibramed–Laserpulse[1] | 830 | >120 | over 36 | <30 | 30.40 | 101% |
| Ibramed–Laserpulse[1] | 660 | >120 | over 36 | <30 | 18.08 | 60% |
| Ibramed–Laserpulse[1] | 660 | >120 | over 36 | <30 | 16.93 | 56% |
| Ibramed–Laserpulse[1] | 904 | >120 | over 36 | <30 | 52.08 | 74% |
| Ibramed–Laserpulse[1] | 660 | >120 | never | <30 | 34.28 | 114% |
| Ibramed–Laserpulse[1] | 660 | >120 | never | <30 | 27.13 | 90% |
| Ibramed–Laserpulse[1] | 904 | >120 | never | <30 | 50.04 | 71% |
| Ibramed–Laserpulse[1] | 904 | >120 | never | <30 | 41.86 | 60% |
| Ibramed–Laserpulse[1] | 904 | >120 | never | <30 | 43.08 | 62% |
| Ibramed–Laserpulse[1] | 830 | >120 | never | <30 | 31.25 | 104% |
| Ibramed—Antares[1] | 660 | <12 | <12 of acquisition | <30 | 42 | 105% |
| Ibramed—Antares[1] | 904 | <12 | <12 of acquisition | <30 | 72.1 | 103% |
| HTM–Fluence[2] | 904 | ±96 | 12 to 24 | <30 | 7.64 | 59% |
| HTM–Fluence[2] | 904 | ±96 | 12 to 24 | 30 to 90 | 11.4 | 88% |
| DMC–Therapy XT[3] | 808 | <12 | <12 of acquisition | 30 to 90 | 95.5 | 96% |
| DMC–Therapy XT[3] | 660 | <12 | <12 of acquisition | 30 to 90 | 103 | 103% |

[1] 2800 Dr. Carlos Burgos Ave, Amparo, São Paulo, Brazil

[2] 831 Sebastião Moraes St, São Carlos, São Paulo, Brazil

[3] 209 Rio Nilo Ave, Amparo, São Paulo, Brazil.

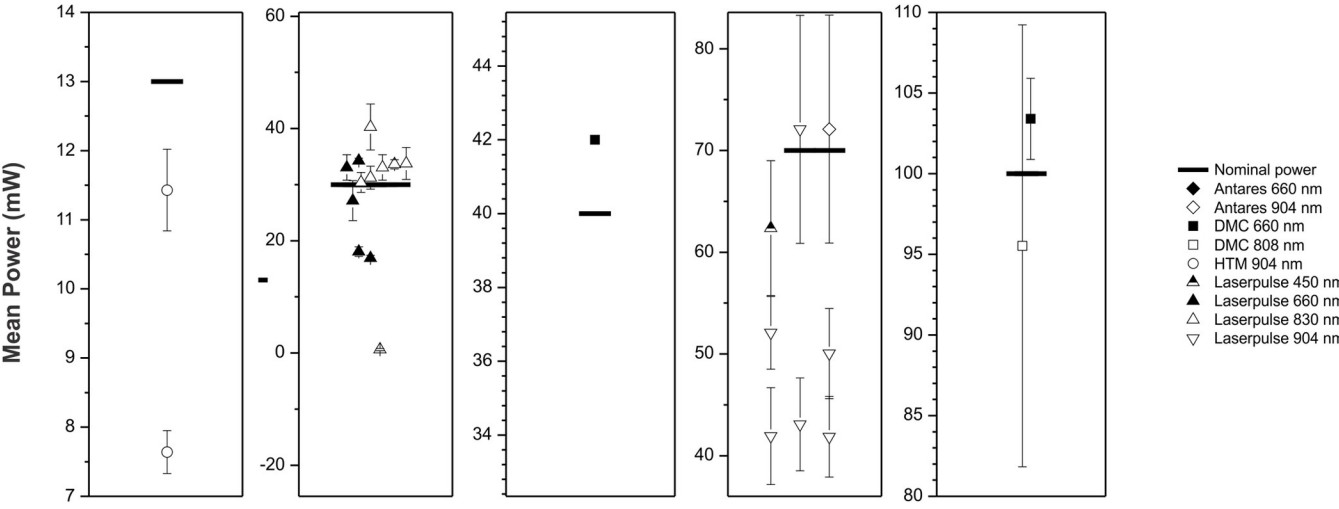

**Fig 2. The average power of different equipment against nominal power.**

The data found for the average power of each piece of equipment is shown in Fig 2, presented as an average and its deviations during one minute of irradiation compared to the value quoted by each manufacturer.

The beam diameter also showed differences between the measured values and those declared in the manufacturer's manual (Fig 3), especially for the same brand and model equipment that showed differences between such results. Thus, the diameters at different distances for the wavelengths 450 nm and 660 nm (Fig 3A), 830 nm, and 808 nm (Fig 3B), as well as for 904 nm (Fig 3C) are highlighted.

Additionally, associating Fig 3 and Table 2, in addition to the analysis of the different divergences found, we can extrapolate to other Z from Eq 3:

$$r = r0 + \tan\frac{\theta}{2} * z - z0 \qquad\qquad \text{Eq3}$$

where *r* is equal to the position on the radius, *r0* is the radius at the tip or probe, *Z* is the distance, and *z0* is the initial distance at the tip, which was considered zero here. Since slope indicates the inclination of a line and intercept indicates where it intersects an axis, they define linearity between two variables and can be employed to estimate an average rate of change.

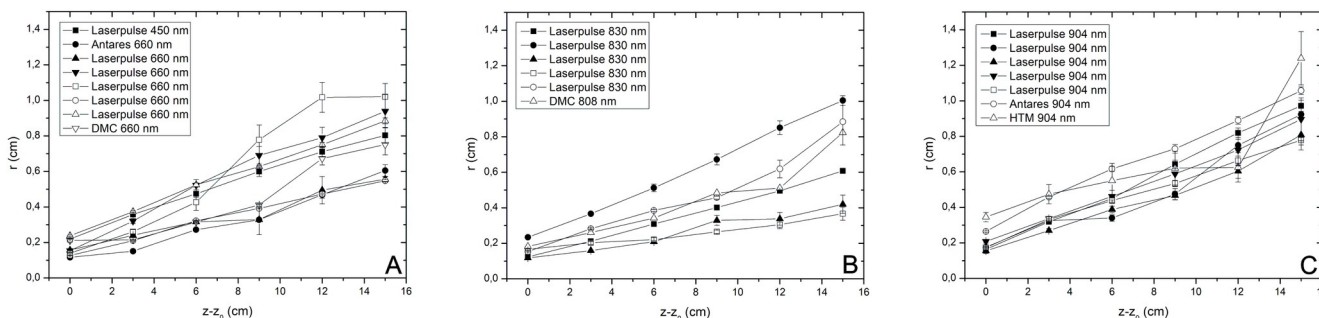

**Fig 3. Laser beam diameters in different equipment and different application distances.** Z-$Z_0$ is the different analysis points (0, 3, 6, 9, 12, and 15 cm) of the laser emitter; r is the radius for the highlighted point.

**Table 2. The intercept and slope for different inclination in evaluated equipment.**

| Equipment | Intercept | Slope |
|---|---|---|
| Ibramed—Laserpulse 450nm [1] | 0.220 (0.002) | 0.042 (0.001) |
| Ibramed—Antares 660nm [1] | 0.091 (0.020) | 0.029 (0.003) |
| Ibramed—Laserpulse 660nm [1] | 0.158 (0.002) | 0.027 (0.000) |
| Ibramed—Laserpulse 660nm [1] | 0.154 (0.004) | 0.056 (0.002) |
| Ibramed—Laserpulse 660nm [1] | 0.136 (0.009) | 0.053 (0.006) |
| Ibramed—Laserpulse 660nm [1] | 0.128 (0.005) | 0.029 (0.001) |
| Ibramed—Laserpulse 660nm [1] | 0.242 (0.005) | 0.043 (0.001) |
| DMC–Therapy XT 660nm [3] | 0.123 (0.018) | 0.031 (0.005) |
| Ibramed—Laserpulse 830nm [1] | 0.119 (0.003) | 0.032 (0.000) |
| Ibramed—Laserpulse 830nm [1] | 0.229 (0.006) | 0.050 (0.002) |
| Ibramed—Laserpulse 830nm [1] | 0.117 (0.002) | 0.015 (0.001) |
| Ibramed—Laserpulse 830nm [1] | 0.166 (0.004) | 0.011 (0.001) |
| Ibramed—Laserpulse 830nm [1] | 0.157 (0.007) | 0.037 (0.002) |
| DMC–Therapy XT 808nm [3] | 0.141 (0.054) | 0.039 (0.006) |
| Ibramed—Laserpulse 904nm [1] | 0.159 (0.013) | 0.052 (0.003) |
| Ibramed—Laserpulse 904nm [1] | 0.175 (0.009) | 0.049 (0.002) |
| Ibramed—Laserpulse 904nm [1] | 0.156 (0.005) | 0.037 (0.001) |
| Ibramed—Laserpulse 904nm [1] | 0.208 (0.002) | 0.043 (0.001) |
| Ibramed—Laserpulse 904nm [1] | 0.186 (0.011) | 0.041 (0.002) |
| Ibramed—Antares 830nm [1] | 0.269 (0.008) | 0.054 (0.002) |
| HTM–Fluence 904nm [2] | 0.346 (0.038) | 0.034 (0.008) |

[1] 2800 Dr. Carlos Burgos Ave, Amparo, São Paulo, Brazil

[2] 831 Sebastião Moraes St, São Carlos, São Paulo, Brazil; O

[3] 209 Rio Nilo Ave, Amparo, São Paulo, Brazil.

So, after applying Eq 3, a straight line can be fitted, and from such an outcome, the value of *r0* and the $\tan\frac{\theta}{2}$ can be observed. Here we can calculate the beam radius with the probe in contact and the divergence of the beam in degrees, compared with the nominal values stated in the user manual.

Finally, having found the value, one can observe from the equation the divergence angulation value and *r* for any place of *Z*.

Three of the beam shapes analyzed to demonstrate the difference between equipment of the same model can be seen in Fig 4. Also, as determined by the analysis system shown in Fig 1, two radius values can be achieved. Firstly, *r1* for the longer radius and even *r2* for the smaller radius. Thus, for the equipment evaluated, the geometry of the beams is more elliptical, not offering a more circular description as sometimes considered.

No statistically significant differences were observed for the correlation analyses between average age and observed diameter ($r_s$ = -0.249; $p$ = 0.277) or mean power values ($r_s$ = -0.051, $p$ = 0.813). Similarly, between the last maintenance and diameter values ($r_s$ = -0.282, $p$ = 0.216) and mean power ($r_s$ = -0.107, $p$ = 0.619), no significant difference was observed.

Finally, to analyze the stability of power output over time for different wavelengths, power output was monitored for one and three minutes of sequential emission at 830 nm and 904 nm. To analyze the difference when comparing the wavelengths, the test was applied on Δ830 and Δ904, that is, the difference between the values analyzed for one and three minutes for each spectrum. Thus, after analyzing the averages of each group, no significant difference was observed between the groups (Table 3).

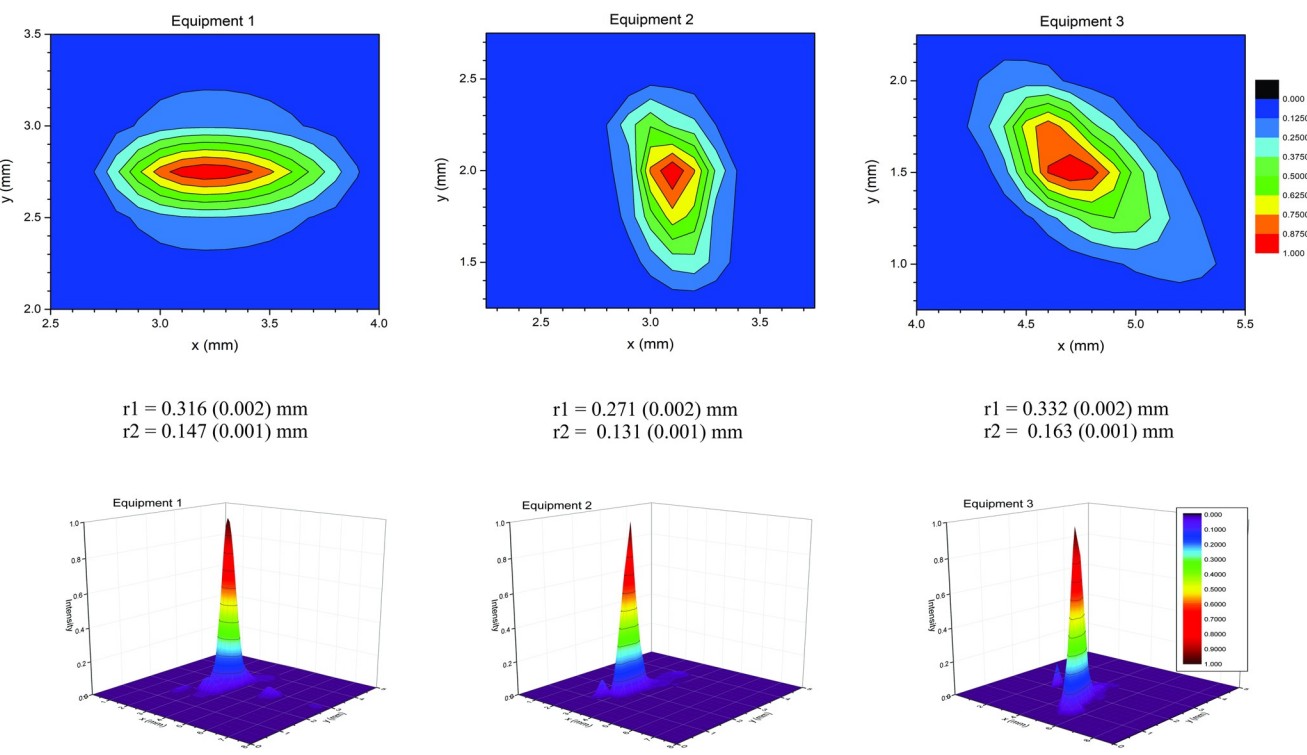

r1 = 0.316 (0.002) mm
r2 = 0.147 (0.001) mm

r1 = 0.271 (0.002) mm
r2 = 0.131 (0.001) mm

r1 = 0.332 (0.002) mm
r2 = 0.163 (0.001) mm

**Fig 4. Beam geometry of three different emitting sources of the same brand and model.**

## Discussion

The objective of this study was to evaluate the consistency of laser beam diameter and the average powers with values quoted in user manuals for several laser light sources. In general, the high degree of variation points to the importance of maintenance and preventive measurements if these light sources are to be used reliably for photomedicine applications. It is noteworthy that such differences may come from the time of manufacture. Thus, the need for evaluation over time is once again highlighted, and it is even recommended in the standards that users perform such checks at appropriate time intervals provided by the manufacturers themselves. Besides, among the development of different therapeutic laser sources, application techniques, and measurement methods, a form of standardization was needed, regardless of

**Table 3. Influence of sequential irradiation time on the average and maximum power evaluated.**

| Outcomes | Power | | Comparison of means |
|---|---|---|---|
| 830–3 Min *vs* 830nm– 1 Min (Maximum) | 31.13 (SD 11.82) | 29.60 (SD 13.14) | $p$ 0.047* |
| 830–3 Min *vs* 830nm– 1 Min (Mean) | 30.20 (SD 11.68) | 28.77 (SD 12.84) | $p$ 0.031* |
| 904–3 Min *vs* 904nm– 1 Min (Maximum) | 41.92 (SD 22.01) | 41.08 (SD 21.88) | $p$ 0.019* |
| 904–3 Min *vs* 904nm– 1 Min (Mean) | 41.09 (SD 21.70) | 40.02 (SD 21.23) | $p$ 0.006* |
| D830 *vs* D904nm (Maximum) | 3.33 (SD 3.33) | 3.25 (SD 3.92) | $p$ 0.673 |
| D830 *vs* D904nm (Mean) | 3.33 (SD 2.88) | 3.25 (SD 2.05) | $p$ 0.865 |

SD: Standard Deviation; $p$: p-value; D830: The difference between 830 (1 Min) and 830 (3 Min); D904: The difference between 904 (1 Min) and 904 (3 Min)

*statistically significant.

the environment, to achieve sustainable levels of quality and safety for both the operator and the patient. So, the implementation of an international standardization was suggested. Wright *et al.* [10] state that the minimum effective beam width, location, and beam divergence as important parameters of laser generating sources. It should also be considered that the beam propagation factor is directly related to its width and divergence.

Different ways of analyzing and quantifying therapeutic light delivery have been proposed. Johnston and Fleischer [11] proposed a new arrangement of the knife-edge method analysis system with absolute accuracy at the level of 0.3% compared to other known analyses. Additionally, different forms of technology have been applied since the early years of the development of the first equipment, has been of equal interest not only for the improvement of the equipment but also for the analysis of the interaction of light and its biological effects and, not least, proper calibration and measurement of the emitting sources [12]. In the 1970s, Miyamoto and Yasuura [13] developed a method for measuring the beam parameters of a laser and its diffraction field using a hologram. Thus, considerable accuracy was already shown, but specific to the He-Ne laser available at the time. Recently, Yang *et al.* [12], Ke *et al.* [14], and Luo *et al.* [15] studied advances and techniques to improve the understanding and analysis of different emitting sources. Therefore, the introduction of new technologies allows the ability to interpret a greater variety of generating sources, exploring the various biophysical parameters or methods imposed, thus inserting new light-emitting technologies, such as diodes, derived from a differentiated laser light emission.

However, although we have the requirements pointed out by the standardization suggested by Wright *et al.* [10], such as the mandatory requirement of the minimum effective beam width, its location, and the beam divergence along with the wavelength and power or fluence, in addition to the technologies imposed for analysis already in stability and reliability processes, some equipment still present discrepancies to the suggested standards. Among the equipment analyzed in our study, all presented such values in their manuals, but one of the models did not show these data in the same space as suggested by the standard but dispersed throughout the text. It should be noted that this point is a suggestion and not mandatory in the Brazilian technical standards. Furthermore, one piece of equipment offered only the diameter value of its optical fiber and not the total area of the emitting source since there are two fibers added to the adjacent space. We also highlight in the operation manuals the presence of reminders for the need for preventive maintenance, where only one brand did not have such a reminder available in its text, where here we highlight the obligation pointed out by the standard. Among those who mention it, it is a consensus that the equipment must be used for 12 months between each maintenance.

The International Electrotechnical Commission gives the standards in their most recent form, and the regulatory standard for light therapy is described under 60825:2020 [9]. For Brazil, the Brazilian Regulatory Standard NBR60601-1-8:2014 applies [7,8], regulated by the Brazilian Association of Technical Standards (ABNT). Thus, according to ABNT NBR IEC 60601-2-22:2014 [7], some conditions must be followed. Firstly, the output of the laser beam should be highlighted, where it should not extrapolate a difference more significant than 20% of the predicted, and it should be explicitly mentioned in the manual. According to subsection 201.12.4.4, the laser output must be checked through a calibrated meter, and it is recommended that the equipment allow continuous verification of light output quantitatively. In this way, we reinforce the obligation of the manufacturer to point out the period between each preventive maintenance, as well as the measuring methods for the equipment. Another point addressed is the beam opening, according to the IEC 60825–1 standard of 2014. Finally, for the safety of the professional and the patient, eye protection tools are reinforced, with a highlighted note provided by the manufacturing company.

The need to follow the rules and knowledge of such variables can be explained by the study of Ash *et al.* [16]. The authors demonstrate applying a simulation in which not only different wavelengths can interfere in the light penetration capacity in biological tissues as presented by Barbosa *et al.* [17], but also the beam diameter as the simulation of Ash *et al.* [16]. Their results indicate that as the beam width increases, there is also an increase in the penetration depth. Furthermore, a method in which the beamwidth changes from 1 to 40 mm were proposed by their group, demonstrating that beam diameters between 1 mm and 5 mm have a notable increase in penetration depth, this increase in penetration is smaller for beams between 5 mm and 12 mm, and this increase stops for higher diameters. This demonstrates the importance of the therapist knowing the beamwidth of the device at his disposal, as well as an accurate description of the companies in front of the construction stage. At the same time, Kwon *et al.* [18] also point out the influence of beam diameter on the penetration capacity of the therapeutic light beam. Also, utilizing mathematical projections, they point out the possibility of association with different methods to improve such penetration capacity. Thus, our findings highlight the need for specific values of imposed diameters.

Some studies point to the need for preventive maintenance of electrophysical equipment used in physical therapy [19–21], highlighting the one that uses photobiomodulation [5,6]. The authors evaluated the average power, characteristics of use, and precautions for preventive maintenance. It should be noted that these studies have been going on for a few years, and the consequent improvement of the technology embedded in the equipment, so far, was not found in any research that evaluated the light beam. Our results indicate that ten devices evaluated were outside the established standards, with nine of them having at least 20% less than the predicted power and as much as 98% less. One device was 20% higher than predicted, precisely 34% more. Thus, two possible causes can be highlighted. Initially, the low application of preventive maintenance can be considered since it is impossible to notice variations in this variable in the clinical environment. However, it is also worth mentioning the possible discrepancy of values in the manufacturing process since certification centers only test samples and not a whole batch of equipment, generating significant differences between the equipment and generating sources.

As highlighted in Fig 4, the difference in the beam geometry of different equipment is notorious, even when comparing within the same manufacturer and model of equipment. However, our study does not allow us to say if this difference is due to differences in manufacturing processes or changes during the use of the equipment in clinical routine since we had no comparison factor between the two conditions. It is known that changing these parameters leads to unrealistic values of fluence and power density, which may interfere in the light penetration depth, as well as the biological response. Another essential point to be discussed is related to the non-reproducibility of some results published in clinical practice scientific reports because in some of these studies, the real power emitted by the equipment has not been measured. Fonseca [22] also argues that there are real difficulties in comparing experimental and clinical results since there are numerous possible interactions of physical and biological parameters that should be checked before irradiation procedures, in addition to the occasional omission or inaccurate reports as pointed out by our findings.

From this, we can point, for example, to the expected energy equation, where *Energy* (*J*) = *Power* (*W*)* *Time* (*s*). In non-measured equipment, not having the correct emission value, the value presented in the manual will be considered, although as observed in our results, the equipment often does not deliver the fluence declared by the manufacturers. Similarly, to consider power density, where *Power density* $(W/cm^2) = \frac{Power\ (W)}{Area\ (cm^2)}$, the area of the emitting source is of total importance for the outcome and also, as our results pointed out, in some cases,

diverges from the one presented in the manual. Finally, another factor observed is the beam conformation, where they are described mainly by cylindrical beams and a perfect Gaussian profile, but as observed, after passing through the lenses or fibers of their generating equipment, they tend to lose or drastically change their shape. Thus, it is important to emphasize again that all these differences will have a significant effect on the penetrative capacity of the light beams and consequent exposure to photons resulting in a significant difference in the treatment outcome.

A limitation presented in the study was the low variability of models, although it expresses the reality of the equipment available in physical therapy services. Also, it should be noted that the equipment was not monitored over time, so the state of equipment at a single point in time was analyzed. Finally, a point to be considered is that the article discusses diameter values and not area, usually observed in scientific papers. This occurred due to the lack of discussion in the equipment manuals regarding elliptical and circular beams, in addition to the data collection covering only one diameter. Thus, the most robust method was the discussion by diameter.

## Conclusion

Our findings point out that there is a need for a more significant shared effort on the part of the manufacturers to describe the specifications of the equipment, making them more accessible to the user and the operators themselves as to care and maintenance. The results show that preventive maintenance of the equipment is not routinely done, which may have resulted in the significant difference between the values measured here and those predicted by the manufacturer in their respective manuals. Ultimately these discrepancies can introduce significant uncertainty into the efficiency and effectiveness of photobiomodulation therapy.

## Author Contributions

**Conceptualization:** Carlos Eduardo Girasol, Luciano Bachmann, Rinaldo Roberto de Jesus Guirro.

**Data curation:** Carlos Eduardo Girasol, Guilherme de Araújo Braz, Luciano Bachmann, Jonathan Celli, Rinaldo Roberto de Jesus Guirro.

**Formal analysis:** Carlos Eduardo Girasol, Guilherme de Araújo Braz, Luciano Bachmann, Jonathan Celli, Rinaldo Roberto de Jesus Guirro.

**Funding acquisition:** Carlos Eduardo Girasol, Rinaldo Roberto de Jesus Guirro.

**Investigation:** Carlos Eduardo Girasol.

**Methodology:** Carlos Eduardo Girasol, Guilherme de Araújo Braz, Luciano Bachmann.

**Supervision:** Rinaldo Roberto de Jesus Guirro.

**Writing – original draft:** Carlos Eduardo Girasol, Guilherme de Araújo Braz, Luciano Bachmann, Jonathan Celli, Rinaldo Roberto de Jesus Guirro.

**Writing – review & editing:** Carlos Eduardo Girasol, Guilherme de Araújo Braz, Luciano Bachmann, Jonathan Celli, Rinaldo Roberto de Jesus Guirro.

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
