## [Decision Letter · Decision Letter 0]

8 Feb 2022

PONE-D-21-39449Laser light sources for Photobiomodulation: The role of power and beam characterization in treatment accuracy and reliabilityPLOS ONE

Dear Dr. Guirro,

Thank you for submitting your manuscript to PLOS ONE. After careful consideration, we feel that it has merit but does not fully meet PLOS ONE’s publication criteria as it currently stands. Therefore, we invite you to submit a revised version of the manuscript that addresses the points raised during the review process.

The reviewers were generally positive, but requested a number of minor revisions

We look forward to receiving your revised manuscript.

Kind regards,

Michael R Hamblin

Academic Editor

PLOS ONE

Journal Requirements:

2. Please clarify in your Methods section how permission was sought to access the equipment, and who provided permission.

Reviewers' comments:

Reviewer's Responses to Questions

**Comments to the Author**

1. Is the manuscript technically sound, and do the data support the conclusions?

Reviewer #1: Yes

Reviewer #2: Yes

2. Has the statistical analysis been performed appropriately and rigorously? 

Reviewer #1: Yes

Reviewer #2: Yes

3. Have the authors made all data underlying the findings in their manuscript fully available?

Reviewer #1: Yes

Reviewer #2: Yes

4. Is the manuscript presented in an intelligible fashion and written in standard English?

Reviewer #1: Yes

Reviewer #2: Yes

5. Review Comments to the Author

Reviewer #1: I thank the Authors for their evaluation of laser photobiomodulation device parameters and detailed discussion thereof.

This manuscript is well-written and comprehensive, and the methodology is reported in sufficient detail to allow reproduction.

The analyses addressed only devices manufactured and used in Brazil, however, the conclusions drawn and recommendations presented have broader applicability, being relevant to devices in use throughout the rest of the world.

I have only a minor alteration and some corrections to suggest:

1. That the study was limited to devices produced and used in Brazil should be stated in the Abstract, perhaps as follows (second sentence, first paragraph): "The aim of this study was to evaluate the average and maximum power and to describe the beam diameter of different low-power laser photobiomodulation devices in clinical use in Brazil.".

2. Line 203 (pg 4), "The average power, fell within 90% to 110%...": Please delete the comma after "power".

3. Line 216 (pg 4), "...as well as for 904 nm (Figure 3C) is highlighted.": Please change "is" to ", are".

4. 2nd para, pg 8, "Here we can calculate the beam radios with the probe in contact...": Please change "radios" to "radius".

Thank you.

Reviewer #2: TITLE: Analysis of low-level laser transmission at wavelengths 660, 830 and 904 nm in biological tissue samples

Page 2 Line 5

The authors report that “Differences in beam

55 diameter of between 38% and 543% of the nominal values”

COMMENT most laser is used for Photobiomodulation are semiconductor/dialysers which typically have elliptical beams in which case they do not have a diameter (strictly speaking they have two diameters)

SUGGESTION describe the differences in the beam area rather than diameter

Page 3 Line 104

The authors report that "Photobiomodulation parameters associated with its effectiveness are not only fluence and light penetration depth but also full optical properties, such as refractive index, absorption, scattering coefficients, beam diameter, and stable output power." also line 115 “However, the equipment manufacturers have predefined some base fluence values without considering the individual characteristics of each subject”

line 116 this context, if the power and the beam diameter are different than expected due to a faulty function of the equipment, the quality of the treatment will be affected

COMMENT the authors mixed photobiomodulation parameters and topics of light propagation into one sentence

light penetration depth refractive index, absorption, scattering coefficients are are topics of light propagation.

I think it is beyond the scope of this paper tooth tackle matters of individual characteristics of each patient and a stick to the key objective which is “ evaluate and describe the

119 laser beam diameter and the average power in different devices in clinical use in Brazil.”

SUGGESTION remove references to light propogation in the patient as follows

114 [2] Enwemeka reports that trained professionals can only control the physical parameters of the laser to be used for each

115 pathology, given its individuality.

116 if the

117 power and the beam diameter are different than expected due to a faulty function of the equipment, the

118 quality of the treatment will be affected. Thus, the objective of this study was to evaluate and describe the

119 laser beam diameter and the average power in different devices in clinical use in Brazil.

I added the word “only” on line 114

Page 4 Line 139-141

The authors report that "The devices with wavelengths in the red spectrum were analyzed

140 during one minute of sequential irradiation and the infrared ones during one and three minutes to observe

141 possible emitted power changes"

COMMENT why did you choose longer treatment times for the infrared lasers ?

SUGGESTION state the reasons why you choose longer treatment times for the infrared lasers

Page 11 Line (no line reference on manuscript)

The authors report that "Their results indicate that as the beam width increases, there is also an increase in the penetration depth. Furthermore, a method in which the beamwidth changes from 1 to 40 mm were proposed by their group, demonstrating that beam diameters between 1 mm and 5 mm have a notable increase in penetration depth"

COMMENT I downloaded the reference paper (Barbosa 2020 Analysis of low-level laser transmission at wavelengths 660, 830 and 904 nm in biological tissue samples. PMID 32516626). The referenced paper does not conclude that as beam width increases, there is also an increase in the penetration depth. It seems both intuativly wrong and my own punpublished experiemnts on pit tissue find the opposite.

SUGGESTION remove all references to increased beam leads to increased penetration.

Page 12 Line (no line reference on manuscript)

The authors report that "the real fluence emitted by the equipment has not been measured"

COMMENT equipment does not emit “fluence”, it emits photons at a wavelength, photon per second equal power, when divided by a beam area on the pataint equals irradiance, and when a doctor, nurse or therapist applies that irradiance for a certain amount of time then a certain fluence has been delivered.

SUGGESTION "the real power or irradiance emitted by the equipment has not been measured"

Page 12 CONCLUSION

I agree with this conclusion. The Photobiomodulation literature is plagued by the problems described.

6. PLOS authors have the option to publish the peer review history of their article (what does this mean?). If published, this will include your full peer review and any attached files.

Reviewer #1: **Yes: **Peter A. Jenkins, M.B.A.

Reviewer #2: **Yes: **James D Carroll

---

## [Author Response · Author response to Decision Letter 0]

17 Feb 2022

- Reviewer #1

Comment: That the study was limited to devices produced and used in Brazil should be stated in the Abstract, perhaps as follows (second sentence, first paragraph): "The aim of this study was to evaluate the average and maximum power and to describe the beam diameter of different low-power laser photobiomodulation devices in clinical use in Brazil.".

Answer: We thank the reviewer. This suggestion has been accepted and highlighted in the manuscript. 

Comment: Line 203 (pg 4), "The average power, fell within 90% to 110%...": Please delete the comma after "power".

Answer: The suggestion was accepted and highlighted in the manuscript. Thank you.

Comment: Line 216 (pg 4), "...as well as for 904 nm (Figure 3C) is highlighted.": Please change "is" to ", are".

Answer: The suggestion was accepted and highlighted in the manuscript. Thank you. 

Comment: 2nd para, pg 8, "Here we can calculate the beam radios with the probe in contact...": Please change "radios" to "radius".

Answer: The suggestion was accepted and highlighted in the manuscript. Thank you.

- Reviewer #2

Comment: Page 2 Line 5 - The authors report that "Differences in beam diameter of between 38% and 543% of the nominal values" - COMMENT most laser is used for Photobiomodulation are semiconductor/dialysers which typically have elliptical beams in which case they do not have a diameter (strictly speaking they have two diameters) - SUGGESTION describe the differences in the beam area rather than diameter.

Answer: The suggestion was considered. However, we cannot state the difference in the area since the nominal definition in the Brazilian devices is given in general averages. Thus, as our data is presented and made available, only one diameter was measured and compared. So, unfortunately, changing to an area would produce non-existent data. Also, we would like to point out that in the session Methods/ Beam diameter evaluation, we explain how this value was reached. Furthermore, we emphasize that we have inserted this as a limitation of the study. We are open to further considerations.

Comment: Page 3 Line 104 - The authors report that "Photobiomodulation parameters associated with its effectiveness are not only fluence and light penetration depth but also full optical properties, such as refractive index, absorption, scattering coefficients, beam diameter, and stable output power." also line 115 "However, the equipment manufacturers have predefined some base fluence values without considering the individual characteristics of each subject". Line 116 this context, if the power and the beam diameter are different than expected due to a faulty function of the equipment, the quality of the treatment will be affected - COMMENT the authors mixed photobiomodulation parameters and topics of light propagation into one sentence light penetration depth refractive index, absorption, scattering coefficients are topics of light propagation. I think it is beyond the scope of this paper tooth tackle matters of individual characteristics of each patient and a stick to the key objective which is "evaluate and describe the laser beam diameter and the average power in different devices in clinical use in Brazil." - SUGGESTION remove references to light propogation in the patient as follows.

Answer: The suggestion was accepted, and the sentence had been deleted from the manuscript. Thank you.

Comment: Enwemeka reports that trained professionals can only control the physical parameters of the laser to be used for each pathology, given its individuality [...] if the power and the beam diameter are different than expected due to a faulty function of the equipment, the quality of the treatment will be affected. Thus, the objective of this study was to evaluate and describe the laser beam diameter and the average power in different devices in clinical use in Brazil. - I added the word "only" on line 114.

Answer: The suggestion was accepted and highlighted in the manuscript. Thank you.

Comment: Page 4 Line 139-141 - The authors report that "The devices with wavelengths in the red spectrum were analyzed during one minute of sequential irradiation and the infrared ones during one and three minutes to observe possible emitted power changes" - COMMENT why did you choose longer treatment times for the infrared lasers? - SUGGESTION state the reasons why you choose longer treatment times for the infrared lasers.

Answer: The suggestion was discussed, resubmitted, and highlighted in the manuscript. Thank you.

Comment: Page 11 Line (no line reference on manuscript) - The authors report that "Their results indicate that as the beam width increases, there is also an increase in the penetration depth. Furthermore, a method in which the beamwidth changes from 1 to 40 mm were proposed by their group, demonstrating that beam diameters between 1 mm and 5 mm have a notable increase in penetration depth" - COMMENT I downloaded the reference paper (Barbosa 2020 Analysis of low-level laser transmission at wavelengths 660, 830 and 904 nm in biological tissue samples. PMID 32516626). The referenced paper does not conclude that as beam width increases, there is also an increase in the penetration depth. It seems both intuativly wrong and my own punpublished experiemnts on pit tissue find the opposite. - SUGGESTION remove all references to increased beam leads to increased penetration.

Answer: We agree that the citation was misplaced. The central idea was in front of the wavelength in biological tissues as presented by Barbosa 2020 and the wavelength of the beam according to the extrapolation of Ash 2017. The text was reorganized for the correct interpretation. Thank you for the suggestion.

Comment: Page 12 Line (no line reference on manuscript) - The authors report that "the real fluence emitted by the equipment has not been measured" - COMMENT equipment does not emit "fluence", it emits photons at a wavelength, photon per second equal power, when divided by a beam area on the pataint equals irradiance, and when a doctor, nurse or therapist applies that irradiance for a certain amount of time then a certain fluence has been delivered. - SUGGESTION "the real power or irradiance emitted by the equipment has not been measured". 

Answer: The suggestion was accepted and highlighted in the manuscript. Thank you.

Comment: Page 12 CONCLUSION - I agree with this conclusion. The Photobiomodulation literature is plagued by the problems described.

Answer: Thank you for your appreciation.

---

## [Editor Report · Decision Letter 1]

16 Mar 2022

Laser light sources for Photobiomodulation: The role of power and beam characterization in treatment accuracy and reliability

PONE-D-21-39449R1

Dear Dr. Guirro,

We’re pleased to inform you that your manuscript has been judged scientifically suitable for publication and will be formally accepted for publication once it meets all outstanding technical requirements.

Kind regards,

Michael R Hamblin

Academic Editor

PLOS ONE
---

## [Editor Report · Acceptance letter]

21 Mar 2022

PONE-D-21-39449R1 

Laser light sources for Photobiomodulation: The role of power and beam characterization in treatment accuracy and reliability 

Dear Dr. Guirro:

I'm pleased to inform you that your manuscript has been deemed suitable for publication in PLOS ONE. Congratulations! Your manuscript is now with our production department. 

Kind regards, 

on behalf of

Dr. Michael R Hamblin 

Academic Editor

PLOS ONE